# Modulation of PRC1 Promotes Anticancer Effects in Pancreatic Cancer

**DOI:** 10.3390/cancers16193310

**Published:** 2024-09-27

**Authors:** Hajin Lee, An-Na Bae, Huiseong Yang, Jae-Ho Lee, Jong Ho Park

**Affiliations:** Department of Anatomy, School of Medicine, Keimyung University, Daegu 42601, Republic of Korea

**Keywords:** PRC1, immune cell, proliferation, pancreatic cancer, cancer prevention

## Abstract

**Simple Summary:**

Protein Regulator of Cytokinesis (PRC) 1 is traditionally recognized for its role in the final physical stage of cell division. However, recent studies have highlighted its significant involvement in various cancers. This study aimed to assess the clinical relevance of PRC1 in pancreatic cancer and its impact on cancer cell characteristics. Based on data from The Cancer Genome Atlas (TCGA) and a cell-based assay, PRC1 has emerged as a clinical marker for pancreatic cancer. These findings suggest that the regulation of PRC1 provides a promising strategy for developing anticancer therapies in pancreatic cancer.

**Abstract:**

**Background**: Pancreatic cancer, while relatively uncommon, is extrapolated to become the second leading cause of cancer-related deaths worldwide. Despite identifying well-known markers like the *KRAS* gene, the exact regulation of pancreatic cancer progression remains elusive. **Methods**: Clinical value of PRC1 was analyzed using bioinformatics database. The role of PRC1 was further evaluated through cell-based assays, including viability, wound healing, and sensitivity with the drug. **Results**: We demonstrate that PRC1 was significantly overexpressed in pancreatic cancer compared to pancreases without cancer, as revealed through human databases and cell lines analysis. Furthermore, high PRC1 expression had a negative correlation with CD4+ T cells, which are crucial for the immune response against cancers. Additionally, PRC1 showed a positive correlation with established pancreatic cancer markers. Silencing PRC1 expression using siRNA significantly inhibited cancer cell proliferation and viability and increased chemotherapeutic drug sensitivity. **Conclusions**: These findings suggest that targeting PRC1 in pancreatic cancer may enhance immune cell infiltration and inhibit cancer cell proliferation, offering a promising avenue for developing anticancer therapies.

## 1. Introduction

Pancreatic ductal adenocarcinoma (PDAC) is an aggressive cancer and is currently the third leading cause of cancer-related death in the United States [1,2]. By 2030, it is projected to become the second-leading cause of cancer-related mortality [1]. Due to the late stage at which most patients are diagnosed, only about 12% survive beyond five years [2,3]. Treatment options for pancreatic cancer are limited, with surgical resection or chemotherapy being the primary methods [4,5]. Specifically, gemcitabine and paclitaxel are commonly used alongside radiotherapy [4]. A hallmark of pancreatic cancer is desmoplasia, characterized by a dense fibrotic stroma and extracellular matrix (ECM) [6]. This fibrotic physical barrier significantly impedes the effectiveness of chemotherapy, making it challenging to achieve substantial anticancer effects in the type of “cold tumor” [5,7,8]. While immunotherapy has shown high efficacy across various cancers, particularly lymphomas, its success in treating pancreatic and other solid tumors has been limited [6,7,9]. Immune checkpoint inhibitors (ICIs) or Chimeric Antigen Receptor (CAR)-T cell therapies have not demonstrated significant efficacy in these cases [10,11]. Hence, there is an urgent need for new targets and treatments for pancreatic cancer.

PRC1 plays a crucial role in cytokinesis, the final physical stage of cell division [12,13,14]. It contains a microtubule-binding (MT) domain, contributing to cell growth [15]. PRC1 expression levels are tightly regulated, peaking during the S and G2/M phases of the cell cycle and decreasing sharply after cell division [13]. The activity of PRC1 is negatively regulated by phosphorylation to prevent premature assembly of midzones in cells [16,17]. PRC1 promotes cancer cell proliferation and has been identified as a potential therapeutic target for several cancers, including lung and ovarian cancer [18,19,20]. However, the role of PRC1 in “cold tumors”, such as pancreatic cancer, has not been fully studied.

In this study, we investigated PRC1 as an anticancer drug candidate for pancreatic cancer. The clinical and prognostic significance of PDAC was analyzed through a big bioinformatics database based on TCGA. Our study has shown the significant differential expression of PRC1 in pancreatic cancer stages and its correlation with immune cell infiltration and other markers of pancreatic cancer. We also confirmed the crucial role of the PRC1 in vitro experiment, including viability, wound healing, and sensitivity, with the drug. Therefore, our study suggests that PRC1 is one of the potential drug targets for pancreatic cancer, and we expect that blocking PRC1 will improve combination therapy with immunotherapeutics.

## 2. Materials and Methods

### 2.1. UALCAN Database Analysis

The University of Alabama at Birmingham Cancer Data Analysis Portal (UALCAN) utilizes data from the TCGA database to evaluate protein-coding gene expression and its impact on patient survival across 33 types of cancer [21]. Analysis of PRC1 was performed in pancreatic cancer based on sample type, patient sex, age, chronic pancreatitis status, nodal metastasis status, cancer stage, and tumor grade by UALCAN. UALCAN is publicly available at http://ualcan.path.uab.edu (accessed on 1 July 2024).

### 2.2. GEPIA Database Analysis

Gene expression profiling interactive analysis (GEPIA) is a web server based on TCGA that analyzes RNA sequencing expression across various cancer types [22]. GEPIA offers interactive and adaptable functions, including distinctive expression analysis, profiling plotting, correlation, patient survival, and more analysis. Analysis of PRC1 was performed with the overall survival and disease-free survival of pancreatic cancer patients using GEPIA. HR denotes Hazard Ratio and TPM means Transcripts Per Million. GEPIA is available at http://gepia.cancer-pku.cn/ (accessed on 1 July 2024).

### 2.3. TIMER Database Analysis

TIMER is a program for the comprehensive analysis of tumor-infiltrating immune cells across various cancer types [23]. This web server provides estimates of immune infiltrates’ abundances using multiple immune deconvolution methods and allows for the dynamic generation of high-quality figures to explore tumor immunological, clinical, and genomic features based on TCGA data in human genes. The relationship between PRC1 gene expression levels and *KRAS* or *TP53* was analyzed in pancreatic cancer using TIMER. Timer database analysis is available at http://timer.cistrome.org/ (accessed on 1 July 2024).

### 2.4. Immunohistochemistry (IHC) Staining Analysis

The Human Protein Atlas (HPA) focuses on the expression profiles in human tissues of genes both on the mRNA and protein levels. The protein expression data from 44 regular human tissue types are obtained through antibody-based protein profiling using conventional and multiplex IHC [24]. Images of IHC-stained tissue samples are manually annotated for staining intensity, fraction of cells stained, less than 25%, 25–75%, or more than 75%, and relevance to each tissue type and subcellular localization (nuclear and/or cytoplasmic/membrane) of the commented-on cell type. To analyze the PRC1 expression levels, IHC images of PRC1 expression in pancreas and pancreatic cancer were obtained from the HPA. The HPA is available at https://www.proteinatlas.org/ (accessed on 1 July 2024).

### 2.5. Cell Culture and siRNA Transfection

WT839 and KPC cells were cultured at 37 °C in DMEM (Thermo Fisher Scientific, Waltham, MA, USA, catalog no. 11995065) supplemented with 10% fetal bovine serum (FBS, Thermo Fisher Scientific, catalog no. 16000044) and 1X penicillin–streptomycin (antibiotics, Welgene, Gyeongsan, Republic of Korea, catalog no. LS20202) (referred to as ‘D10’). PanC02 cells were incubated at 37 °C in RPMI Medium 1640 (Welgene, catalog no. LM01103) supplemented with the same FBS and antibiotics (referred to as ‘R10’). siRNA transfection in PanC02 cells or KPC was performed using Lipofectamine RNAiMAX (Thermo Fisher Scientific, catalog no. 13778150) according to the manufacturer’s instructions. Scramble siRNA were obtained from Qiagen (Hilden, Germany, catalog no. 1027280), and PRC1 siRNA constructs were obtained from Thermo Scientific (ID: s107541, catalog no, 4390771, ID: s107543, catalog no. 4390771).

### 2.6. Cell Viability Assay

KPC or PanC02 cells were seeded in 12-well plates. After PRC1 knockdown with specific siRNA or scramble siRNA for 24 to 72 h, cells were washed with PBS. Samples were incubated with 5 mg/mL thiazolyl blue tetrazolium bromide (MTT) (Biobasic, Markham ON, Canada catalog no. 298931) solution for 4 h. The viable cells were added with 700 µL dimethyl sulfoxide (DMSO) (Merck, Darmstadt, Germany, catalog no. D2650100ML). Cell viability was analyzed at a wavelength of 570 nm using an Infinite M200 Pro (Tecan, Zürich, Switzerland). Each well in 12-well plates was transferred to 96-well plates for triplicate reading. Three independent experiments were performed in triplicate to generate statistical analysis.

### 2.7. Western Blot

Cells were collected with Pierce RIPA buffer (Thermo Scientific, catalog no. 89900) with 1 × protease inhibitor (Thermo scientific, catalog no. 78440). Lysates were centrifugated at 17,000× *g* rpm for 30 min for further cell lysis. Subsequently, checking the protein concentration in each sample, the same amounts of total proteins were loaded onto our manual-made 10% gels with 1 × Tris/Glycine/SDS buffer (BIO-RAD, Hercules, CA, USA, catalog no.1610772). After the proteins were separated by SDS-Polyacrylamide gel electrophoresis (PAGE), they were transferred to PVDF Western blotting membrane (Roche, Basel, Switzerland, catalog no. 03010040001) with Tris-Glycine Transfer buffer (iNtRON Biotechnology, Seongnam, Republic of Korea, catalog no. IBSBT0291). For the blocking step, the samples were incubated with 5% skim milk (Kisan Bio, Seoul, Republic of Korea, catalog no. MBS1667) in 1 × Tris-Buffered Saline (DYNE Bio, Seongnam, Republic of Korea, catalog no. CBT3085) containing 0.1% TWEEN, called TBS-T for 30 min. Thereafter washing with TBS-T three times, the membranes were subjected to immunoblot with PRC1 and GAPDH antibodies (PRC1 antibody from Cell signaling technology, Danvers, MA, USA, catalog no. 3639S, GAPDH antibody from Cell signaling technology, catalog no. 5174S) in 3% Albumin Bovine (Generay, Shanghai, China, catalog no.9048-46-8) with TBS-T solution overnight at 4 °C. The following day, the membranes were incubated with Rabbit secondary antibody (Jackson ImmunoResearch, West Grove, PA, USA, catalog no. 111035003) after washing with TBS-T. Membranes were incubated with a Western Bright ECL HRP substrate kit (Advansta, San Jose, CA, USA, catalog no. K-12045-D50). All first antibodies were used at a 1:1000 dilution ratio, and Rabbit secondary antibody was used at a 1:5000 dilution ratio.

### 2.8. Wound Healing Assay

KPC or PanC02 cells were seeded in 6-well plates. Following PRC1 knockdown with siRNA, a scratch was made in the center of each well using pipette tips. The cells were washed with PBS and incubated in 2 mL of the R10 medium for PanC02 and D10 medium for KPC. Cell recovery rates were assessed at 0, 24, and 48 h by using the “iSolution Lite ×64” program. Recovery rates were compared across the different time points.

### 2.9. Annexin V Staining Assay

After PRC1 knockdown with siRNA, PanC02 cells were treated with 2 µM doxorubicin. The following day, all medium and cells were collected using a scraper with PBS. The cells were centrifuged at 1500× *g* rpm for 5 min. Samples were resuspended with 1 × binding buffer (BD Bioscience, Franklin Lakes, NJ, USA, catalog no. 5166121E). The samples were stained with FITC annexin V staining buffer and PI buffer (BD Bioscience, catalog no. 556547) for 15 min at room temperature. Samples were analyzed by using a Cytoflex S (Beckman Coulter, Brea, CA, USA).

### 2.10. Statistical Analysis

Statistical differences among the three groups were analyzed using a one-way ANOVA (Section 3.3). An unpaired *t*-test was used to analyze the significance difference for protein band intensity, cell viability, and annexin V staining result measurements. *p*-value < 0.05 is considered significant. Bar graphs show mean + SD.

## 3. Results

### 3.1. PRC1 Is Significantly Overexpressed in Pancreatic Cancer Patients and Has Potential Clinical Significance

To assess the clinical relevance of PRC1 in pancreatic cancer, we analyzed data from TCGA focusing on gene expression and survival rates. PRC1 expression was markedly higher in pancreatic cancer patients compared to the cohort without cancer (*p* = 0.0001) (Figure 1A). Additionally, patients with high PRC1 expression exhibited a reduction in both overall survival (Logrank *p* = 0.0013, *p*(HR) = 0.0016) and disease-free survival rates (Logrank *p* = 0.023, *p*(HR) = 0.026) compared to patients with low PRC1 expression (Figure 1B). These findings suggest that high PRC1 expression is associated with poor prognosis in pancreatic cancer. To further validate the clinical significance of PRC1, we utilized the UALCAN database, analyzing PRC1 expression across different demographics and clinical factors, such as gender, age, metastatic status, cancer stage, and cancer grade. No significant difference in PRC1 expression was observed between genders (Figure 2A), but PRC1 expression varied significantly with age (Figure 2B). Pancreatitis is one of the risk factors for pancreatic cancer, but other factors still contribute to its development (Figure 2C). PRC1 expression was notably higher in patients with nodal metastasis at stage N1 compared to the cohort without cancer (Figure 2D). In addition, PRC1 levels were significantly elevated in stage 2 pancreatic cancer patients compared to the cohort without cancer (Figure 2E), though there was no significant difference in expression among different tumor grades (Figure 2F). These results highlight the clinical importance of PRC1 as a potential biomarker in pancreatic cancer.

### 3.2. PRC1 May Be Related to Immune Cell Infiltration and Other PDAC Markers

Pancreatic cancer is typically classified as a “cold” tumor, which has a low response rate to immunotherapy [25,26,27]. Moreover, pancreatic cancer has a high heterogeneity tumor microenvironment, which can increase resistance to immunotherapy and reduce the infiltration of immune cells [28]. Various approaches have been explored to enhance the efficacy of immunotherapy, including combination therapies and strategies to increase tumor immunogenicity. The immune cell infiltration number, especially T cells within the tumor, is one of the measurements of immunogenicity. We examined the relationship between PRC1 expression and immune cell infiltration in pancreatic cancer using the TIMER database. Our analysis revealed that PRC1 expression does not significantly correlate with CD8+ T cells (R = −0.032, *p* = 0.68), macrophages (R = −0.046, *p* = 0.554), neutrophils (R = 0.118, *p* = 0.125), or dendritic cells (R = 0.137, *p* = 0.0736) (Figure 3). However, PRC1 was found to have a weakly negative correlation with CD4+ T cells (R = −0.228, *p* = 0.00273). Interestingly, PRC1 has a positive correlation with B cells (R = 0.275, *p* = 0.000296) but a weak correlation. These findings suggest that PRC1 regulates immune cell infiltration within the tumor microenvironment, especially CD4+ T cells. Therefore, targeting PRC1 may enhance the efficacy of immunotherapy in pancreatic cancer by modifying the immune landscape of the tumor. Several key markers are associated with pancreatic cancer, including *KRAS*, *TP53*, and *SMAD4* [29,30,31]. We investigated the correlation between PRC1 and these markers in pancreatic cancer. Our analysis revealed that PRC1 is positively correlated with *KRAS* (R = 0.461, *p* < 0.01) and *TP53* (R = 0.186, *p* < 0.05) (Figure 4A,B). These findings suggest that PRC1 may be an additional target gene in pancreatic cancer.

### 3.3. Reduced PRC1 Expression Inhibits the Proliferation of Pancreatic Cancer Cells

To assess PRC1 expression in pancreatic cancer tissues and cells, we analyzed data from the HPA database using IHC. Also, we examined protein expression levels in cell lines derived from mice. IHC images from the HPA indicated that PRC1 protein expression was higher in PDAC than in pancreases without cancer (Figure 5). Additionally, PRC1 expression was lower in WT839, a normal pancreas cell line from wild-type C57BL/6 mice, compared to pancreatic cancer cell lines KPC and PanC02 (Figure 6A). To investigate the functional role of PRC1 in pancreatic cancer, we knocked down PRC1 to assess cell viability and proliferation in both KPC and PanC02 cell lines. siRNA-mediated knockdown of PRC1 significantly reduced its expression in both cell lines (Figure 6B,C). In KPC cells, PRC1 knockdown led to a marked reduction in cell viability at 24, 48 and 72 h compared to the control (Figure 6D). No significant differences in cell viability were observed between the control and PRC1 knockdown groups in PanC02 cell lines at 24 and 48 h. However, a significant reduction in cell viability was noted in the PRC1 knockdown group at 72 h compared to the control (Figure 6E). The difference in PRC1 expression across cell lines resulted in varying viability rates at 24 and 48 h. Additionally, we examined the effect of PRC1 knockdown on cell proliferation using a wound healing assay. While the control group showed proper cell recovery in the scratched area over time in both cell lines, the PRC1 knockdown cells exhibited no recovery at 48 h in both cell lines (Figure 6F–K). These findings support the conclusion that blocking expression of PRC1 impedes cell viability and proliferation in pancreatic cancer.

### 3.4. Blocking of PRC1 Increases Sensitivity to a Chemotherapeutic Drug in Pancreatic Cancer

Pancreatic cancer is notoriously resistant to various chemotherapeutic agents, including doxorubicin (Doxo) and gemcitabine [32,33,34]. We assessed apoptosis in these cells to determine whether inhibiting PRC1 can enhance the sensitivity of pancreatic cancer cells to Doxo. Our results showed that increased Doxo sensitivity in the PRC1 knockdown group was observed compared to the control group, as evidenced by increased levels of apoptosis (Figure 7A,B). These findings suggest that PRC1 is crucial in mediating drug resistance in pancreatic cancer. Therefore, targeting PRC1 can be a promising strategy to enhance the efficacy of chemotherapy in pancreatic cancer patients.

## 4. Discussion

Our research reveals a crucial link between PRC1 and the development of pancreatic cancer. PRC1 has significant clinical implications, including its distinct expression levels across different cancer stages, association with immune cell infiltration, and correlation with established PDAC markers. Furthermore, we found that reducing PRC1 expression effectively inhibits the proliferation and recovery of pancreatic cancer cells. Additionally, blocking PRC1 expression increases the sensitivity of these cells to a chemotherapy drug. Our study suggests that PRC1 could serve as a pancreatic cancer biomarker and a potential drug target.

The *KRAS* gene is a well-established key marker for developing various cancers, including colorectal, lung, and pancreatic cancers [35,36,37]. In particular, approximately 90% of pancreatic cancer patients harbor *KRAS* mutations [36,38,39]. This mutation often initiates the precancerous stage known as pancreatic intraepithelial neoplasia (PanIN), which can progress to PDAC [40]. PRC1 is involved in cell division and proliferation, suggesting that the co-expression of KRAS with PRC1 may play a role in initiating PanIN, as indicated by our correlation findings between these genes. To further investigate the role of PRC1 in pancreatic cancer, a genetic mouse model, such as *Kras* (G12D) with *Tp53* mutations (KPC model), incorporating PRC1 alterations could be valuable for confirming the involvement of PRC1 in the disease’s progression. Additionally, previous studies have demonstrated that PRC1 is associated with other key factors, including Wnt/β-catenin, polo-like kinase 1 (PLK1), and the p21/p27 family in various cancers [19,41,42]. Therefore, further studies are necessary to evaluate the PRC1-mediated pathway with other key factors in pancreatic cancer.

CD4+ T cells are crucial in creating an immunogenic environment in various cancers [43,44]. They encompass several subpopulations, including T helper (Th) 1, Th2, Th17, and regulatory T cells (Tregs) [45,46]. The primary function of CD4+ T cells is to assist other immune cells, such as cytotoxic CD8+ T cells, and to facilitate antibody responses [46,47]. They also regulate innate immune cells by secreting cytokines like interferon (IFN)-γ and Tumor Necrosis Factor (TNF)-α [48,49]. Additionally, CD4+ T cells can exhibit cytotoxic functions, directly leading to cancer cell death [50,51]. This dual functionality, helper and cytotoxic, makes CD4+ T cells a focus in enhancing the effectiveness of immunotherapy [46,52]. Moreover, CAR-CD4 T cells have been explored in cancer immunotherapy [53,54,55]. Immune cell infiltration, including CD4+ T cells, is rarely observed in cold tumors due to the immunosuppressive environment of pancreatic cancer [25,26,27]. This challenge is well known in pancreatic cancer research. However, our study found a weakly negative correlation between PRC1 expression and CD4+ T cell infiltration. This suggests that targeting PRC1 might help overcome the immunosuppressive environment by promoting the recruitment and activity of CD4+ T cells, potentially enhancing the efficacy of immunotherapy.

Pancreatic cancer is known for its resistance to both immunotherapy and chemotherapy [56,57]. Moreover, previous studies have indicated that PRC1 lacks clear clinical value in pancreatic cancer [58]. However, our study demonstrates that reducing PRC1 expression alters cancer cell characteristics and increases sensitivity to Doxo. Currently, there are no small molecule inhibitors or peptides targeting PRC1. Developing such targeted therapies, specifically for pancreatic cancer cells, could be highly beneficial. Combining PRC1 inhibition with chemotherapy (e.g., Doxo) or immunotherapy (e.g., anti-PD-L1) may improve treatment outcomes.

## 5. Conclusions

PRC1 was overexpressed in pancreatic cancer and negatively correlated with CD4+ T cells. The overexpression of PRC1 was related to cell proliferation and drug resistance in cancer cells, resulting in poor prognosis. These findings provide evidence of the proliferation mechanism and environment of cancer development. So, our study suggests that PRC1 is a promising therapeutic target for pancreatic cancer.

## Figures and Tables

**Figure 1 cancers-16-03310-f001:**
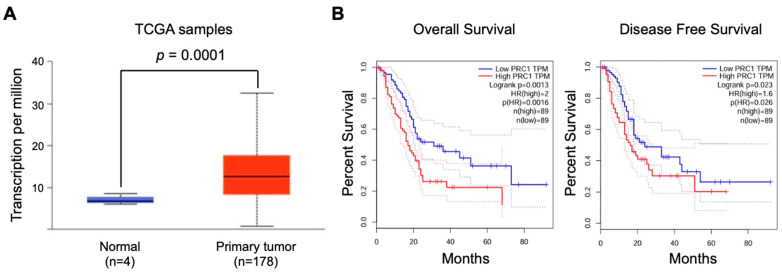
Prognostic significance of PRC1 expression in pancreatic cancer. (**A**) Comparison of PRC1 expression levels in regular pancreases and those with pancreatic cancer. (**B**) Overall survival and disease-free survival rates of pancreatic cancer patients based on high (top 30%) and low (bottom 30%) PRC1 expression levels from the TCGA database (log-rank test; gene expression profiling interactive analysis database). Red and blue dash lines are the 95% confidence interval of each Kaplan-Meier curve.

**Figure 2 cancers-16-03310-f002:**
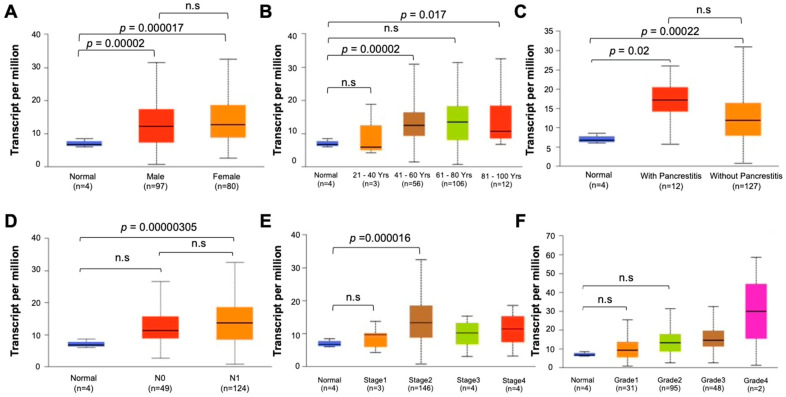
Clinical relevance of PRC1 in pancreatic cancer. (**A**) Sex, (**B**) age, (**C**) histological subtypes with or without pancreatitis, (**D**) nodular metastasis, (**E**) stage, (**F**) grade.

**Figure 3 cancers-16-03310-f003:**
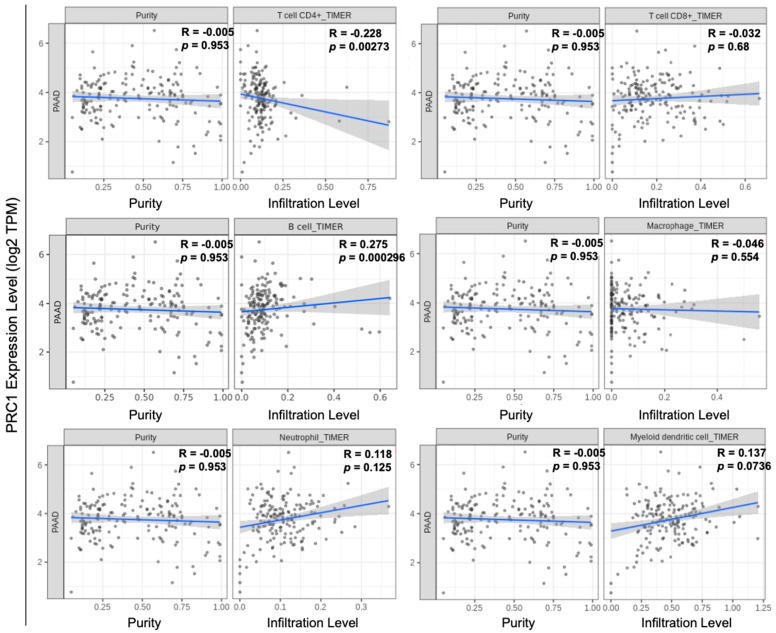
Correlation between PRC1 expression and immune cell infiltration in pancreatic cancer. Analysis of the correlation between PRC1 and the infiltration levels of various immune cells, including CD4+ T cells, CD8+ T cells, B cells, macrophages, neutrophils, and dendritic cells using the TIMER database. The gray dots are immune cells in pancreatic cancer. The gray shadow is 95% confidence interval of each linear regression. The blue line represents the actual measured values for each of gray dots.

**Figure 4 cancers-16-03310-f004:**
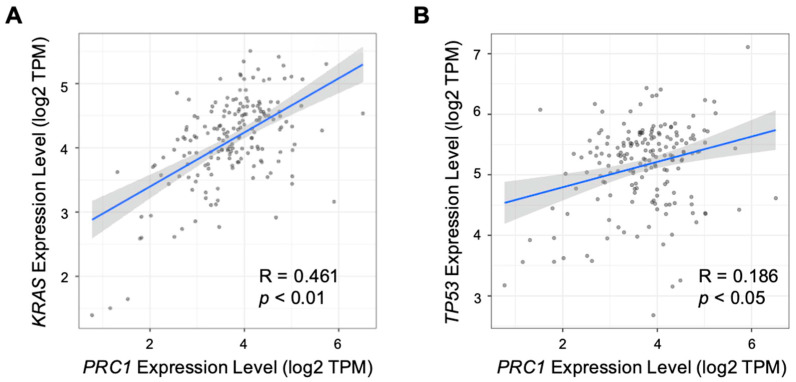
Correlation between PRC1 expression and other markers of pancreatic cancer. Correlation analysis between expression of PRC1 and markers of pancreatic cancer, including (**A**) *KRAS* and (**B**) *TP53*. The gray dots are pancreatic cancer cells. The gray show is 95% confidence interval of each linear regression. The blue line represents the actual measured values for each of gray dots.

**Figure 5 cancers-16-03310-f005:**
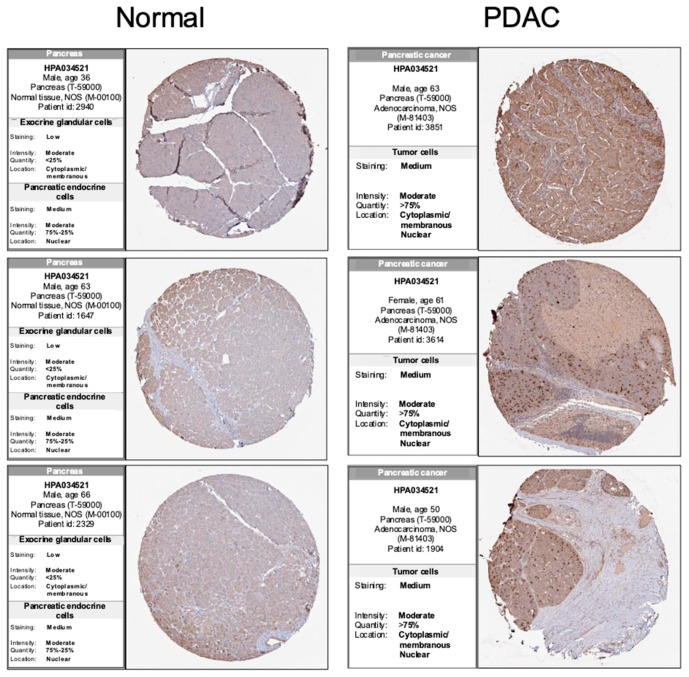
Immunohistochemical analysis of PRC1 expression in pancreatic cancer. Comparison of PRC1 expression in regular pancreases and pancreatic cancer tissue using the HPA database. All images from HPA and IHC scores are ‘enhanced’. Normal tissue IHC images from https://www.proteinatlas.org/ENSG00000198901-PRC1/tissue/pancreas (accessed on 1 July 2024). PDAC tissue IHC images from https://www.proteinatlas.org/ENSG00000198901-PRC1/pathology/pancreatic+cancer#Quantity (accessed on 1 July 2024).

**Figure 6 cancers-16-03310-f006:**
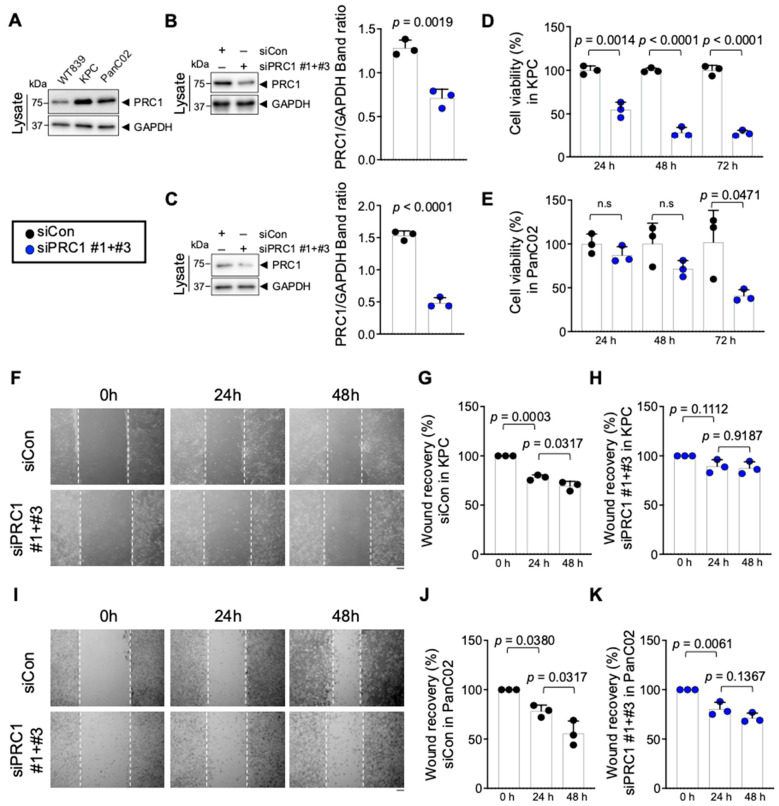
Effect of PRC1 expression on pancreatic cancer cell lines growth inhibition. (**A**) Protein expression of PRC1 in both normal and pancreatic cancer cell lines. (**B**) (Left) Knockdown of PRC1 protein level in KPC cell lines. (Right) The ratio of PRC1/GAPDH protein band intensity was quantified from the three blots shown on the left with two more experiments (*n* = 3). (**C**) (Left) Knockdown of PRC1 protein level in PanC02 cell lines. (Right) The ratio of PRC1/GAPDH protein band intensity was quantified from the three blots shown on the left with two more experiments (*n* = 3). (**D**) Measurement of the viability of KPC cells with PRC1 expression at different time points (*n* = 3). (**E**) Measurement of the viability of PanC02 cells with PRC1 expression at different time points (*n* = 3). (**F**) Representative images of wound healing assay of KPC cell lines with PRC1 expression at different time points. Measurements of wound healing recovery rates in (**G**) control cells group (*n* = 3) and (**H**) PRC1 knockdown KPC cells group (*n* = 3). (**I**) Representative images of wound healing assay of PanC02 cell lines with PRC1 expression at different time points. Measurements of wound healing recovery rates in (**J**) control cells group (*n* = 3) and (**K**) PRC1 knockdown cells group (*n* = 3). Graphs show mean + SD, n.s; not significant, *p* < 0.05: significant. All Western blot images’ raw data were provided in the Appendix A. For (**B**–**E**), unpaired *t*-test. For (**G**,**H**,**J**,**K**), one-way ANOVA and Tukey’s multiple comparisons test. Scale bar: 100 μm.

**Figure 7 cancers-16-03310-f007:**
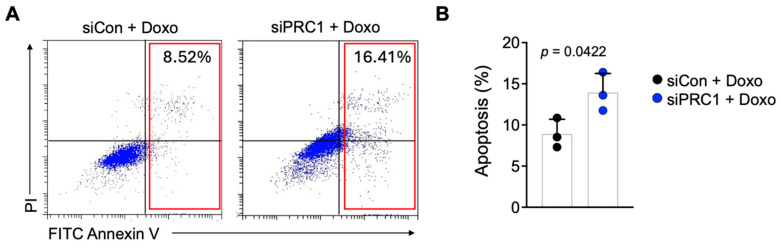
Sensitivity of pancreatic cancer cells to doxorubicin. (**A**) Representative flow plots of Annexin V staining of PanC02 cells after cells were treated with doxorubicin in two groups. (**B**) Measurement of apoptosis rates between control and PRC1 knockdown group (*n* = 3). Graphs show mean + SD, *p* < 0.05: significant. Unpaired *t*-test.

## Data Availability

All results from public datasets were listed in each method section. Full Western blots were added to the Appendix A.

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
