# Peer review of "Modulation of PRC1 Promotes Anticancer Effects in Pancreatic Cancer"

_cancers, 2024, doi:10.3390/cancers16193310_

Round 1
Reviewer 1 Report
Comments and Suggestions for Authors
Park et al. investigated the expression of Protein Regulator of Cytokinesis (PRC) 1 in pancreatic cancer. The authors conducted an analysis of human databases and cell lines, revealing a significantly higher PRC1 expression in pancreatic cancer tissues compared to normal tissues. Moreover, high PRC1 expression was found to be inversely correlated with CD4+ T cells, which play a crucial role in the immune response against cancer. Additionally, PRC1 showed a positive correlation with established pancreatic cancer markers.
Through the use of siRNA to silence PRC1 expression, the authors observed a significant inhibition of cancer cell proliferation, survival ability, and sensitivity to chemotherapeutic drugs. These findings suggest that targeting PRC1 in pancreatic cancer may enhance immune cell infiltration and inhibit cancer cell proliferation, offering a promising avenue for developing effective anti-cancer therapies.
Upon literature review, it was evident that no prior research had elucidated the role of PRC1 in anti-pancreatic cancer. Therefore, this study is deemed to possess certain innovative and valuable aspects. Nevertheless, further experimentation is warranted to comprehensively explore this topic. In light of the aforementioned points, I am inclined to recommend acceptance of the manuscript following minor revisions.
The following is suggestion:
1. The author should meticulously scrutinize the complete manuscript, ensuring proper spacing between numbers and units. Certain characters ought to be formatted in italics or bold, while initial explanations for abbreviations must be provided and subsequently used in their abbreviated form.
2. An additional histogram for WB is required in Figure 6. Furthermore, the author should ensure that n = 3 for both Figure 6A and Figure 6B, and a note should be included in the figure legend to clarify this.
3. Ref1 provides a detailed account of the tumor microenvironment in pancreatic cancer, which can be cited after ‘Pancreatic cancer is typically classified as a "cold" tumor, which has a low response rate to immunotherapy [25-27].’
[Ref1] A. Gu, J. Li, S. Qiu, S. Hao, Z.-Y. Yue, S. Zhai, M.-Y. Li, Y. Liu, Pancreatic cancer environment: From patient-derived models to single-cell omics. Mol. Omics 2024, 20, 220–233. DOI: 10.1039/D3MO00250K
4. The author needs to carefully check the reference format to ensure it meets the requirements of the journal.
5. The iThenticate report indicates that the manuscript exhibits a 41% duplication rate, necessitating further reduction by the author.
Comments on the Quality of English LanguageThe article needs to have its duplication rate reduced.
Reviewer 2 Report
Comments and Suggestions for Authors
Overall, the article is very much significant as it is important to study the role of PRC1 in pancreatic cancer.
Following are a few important points to be considered by authors:
1)Please mention the rationale behind choosing specifically PRC1. Did the authors do any type of bioinformatic analysis and chose this protein among others?
2) Please go through the citation guidelines carefully for each databases such as TCGA, HPA etc. and follow the instructions as mentioned.
3) Authors have used mouse pancreatic cancer cell lines for their in vitro experiments. Why didn't the authors consider using human pancreatic cancer cell lines instead? Please explain.
4) In the methods section, Authors may elaborate in detail about the cell viability, wound healing and Annexin V staining. The methodology has to be much more clearer to the reader. It not clear how may replicates were performed in each experiment and if the authors performed independent experiments to verify and confirm the results.
5) Please follow the guidelines given by HPA when using the images from their study.
6)Discussion needs to be improved by comparing or correlating the results from other studies with PRC1 and other cancers.
7) Recent research study on PRC1 in pancreatic cancer (Vojtech Hanicinec et.al., Oncol Lett. 2021 Aug; 22(2): 598) was not cited in the article. This study could be cited as it states that PRC1 has no prognostic role in colorectal and pancreatic cancer.
8)A few more comments for authors are included in the attached PDF.

Reviewer 3 Report
Comments and Suggestions for Authors The article under review is dedicated to investigation of anti-cancer properties of PRC1 protein. Authors claim that this protein mediates growth and migration of pancreatic cancer cells and confirm that using bioinformatic analysis and in vitro experiments.The article is concise, well-written, easy to follow
Introduction provides necessary information about the research topic
Methods should be supplemented
Research design is appropriate, however provided data should be clarified/supplemented
Author conclusions are partially confirmed by experimental results
The discussion provides necessary comparison with previous works and outlines future research prospects
My points are:
1. Methods 2.3. – describe the analysis in more detail. Human or mouse genes were analyzed?
2. Methods 2.6 – provide the information about scramble RNA.
3. Methods 2.7 – provide gating strategy (singlet isolation) and/or some positive controls (staurosporine) for Annexin V assay.
4. For all ANOVAs provide post hoc test description in Figs legend.
5. Show individual points on Figs 1 and 2.
6. Please compare your up-regulations with those for known biomarkers. E.g. in Fig 1D the PRC1 is up-regulated from ~ 8 to ~12 or ~14 TPMs – can you compare that up-regulation with such of EGFR, another molecule implicated in PDAC progression (10.2147/IJN.S226628)? Is your up-regulation larger or smaller? How many cases of PDAC with mutations are known? You showed the correlation of PRC and KRAS EXPRESSION but does PRC1 expression linked with KRAS MUTATIONAL STATUS? The same for TP53 and SMAD4.
7. Please clarify what does your correlation coefficients mean. The R like -0.28 (Fig 3, consider also big mistakes in high expression values), 0.46 (Fig 4A) are small and I suppose barely have any physiological relevance. Can you somehow advocate this?
8. Please clarify human or mouse genes were compared in TIMER base?
9. Fig 5 – were the conditions of ICH the same? HPA has some protein score can you provide this? Also, please resolve the issue with copyright (at least provide link to every image).
10. Your experiments are trustworthy but should be replicated on other PDAC cells to be sure that the results have relevance.
11. How did you distinguish cell proliferation from migration in scratch assay on Fig 6?
12. It seems that in Fig 7 the necrotic cell population appears, so can PRC1 knockdown cause necrosis?
Generally, the article is interesting and can provide new information about PRC1 as marker of PDAC but is should be revised: correlation analysis should be reinterpreted, experiments should be replicated on other cell lines, some minor issues (italicize in vitro on line 56 and so on) should be corrected
I suggest major revision.
Round 2
Reviewer 2 Report
Comments and Suggestions for Authors
Authors have addressed the reviewers' comments.
Author Response
We thank the reviewer for their constructive comment.
Reviewer 3 Report
Comments and Suggestions for Authors
The authors answered properly on majority of my comments, however, I still have some comments:
1. Methods:
a) Add Abs dilutions in WB part
b) Clarify whether did you eliminated FCS in scratch assay. Did you use some proliferation inhibitors?
2 (prev 7). Your correlation analysis (Fig 3, R = -0.228 and 0.275) should be interpreted with caution. I suggest you to change 3.2. heading to “PCR1 MAY BE related to immune cell…”. You may also add some phrase about your study limitations in discussion.
3 (prev 10). I advise you to replicate your experiments on other PDAC cells to be sure that the results have relevance. That point is important because the article opens up space for new anti-PDAC therapeutic strategies.
I appreciate authors’ research and suppose that revision should be done.
